# Regulation of Rho GTPases by RhoGDIs in Human Cancers

**DOI:** 10.3390/cells8091037

**Published:** 2019-09-05

**Authors:** Hee Jun Cho, Jong-Tae Kim, Kyoung Eun Baek, Bo-Yeon Kim, Hee Gu Lee

**Affiliations:** 1Immunotherapy Research Center, Korea Research Institute of Bioscience and Biotechnology, Daejeon 34141, Korea; hjcho@kribb.re.kr (H.J.C.); kjtdna@kribb.re.kr (J.-T.K.); kebaek@kribb.re.kr (K.E.B.); 2Anticancer Cancer Research Center, Korea Research Institute of Bioscience and Biotechnology, Cheongju 28116, Korea; 3Department of Biomolecular Science, University of Science and Technology (UST), Daejeon 34141, Korea

**Keywords:** RhoGDI1, RhoGDI2, Rho GTPases, cancer, migration, metastasis

## Abstract

Rho GDP dissociation inhibitors (RhoGDIs) play important roles in various cellular processes, including cell migration, adhesion, and proliferation, by regulating the functions of the Rho GTPase family. Dissociation of Rho GTPases from RhoGDIs is necessary for their spatiotemporal activation and is dynamically regulated by several mechanisms, such as phosphorylation, sumoylation, and protein interaction. The expression of RhoGDIs has changed in many human cancers and become associated with the malignant phenotype, including migration, invasion, metastasis, and resistance to anticancer agents. Here, we review how RhoGDIs control the function of Rho GTPases by regulating their spatiotemporal activity and describe the regulatory mechanisms of the dissociation of Rho GTPases from RhoGDIs. We also discuss the role of RhoGDIs in cancer progression and their potential uses for therapeutic intervention.

## 1. Introduction

Rho GTPases regulate various cellular processes, including cell motility, cell adhesion, cytokinesis, cell polarity, cell cycle, and cell survival [1,2]. Anomalous signaling of Rho GTPases is commonly found in many human cancers and can be attributed to several mechanisms, such as overexpression of Rho GTPases with oncogenic activity or alterations of upstream regulators or downstream effectors [3,4]. As in the Ras superfamily, Rho GTPases cycle between an active guanosine triphosphate (GTP)-bound firm in the cell membrane and an inactive guanosine diphosphate (GDP)-bound in the cytoplasm [5]. This cycling is highly regulated by three classes of proteins. Rho guanine nucleotide exchange factors (RhoGEFs) promotes the exchange of GDP for GTP, thereby activating Rho GTPases [6]. Rho GTPase-activating proteins (RhoGAPs) catalyze intrinsic GTP hydrolysis, thereby inactivating Rho GTPases [7]. Rho-specific guanosine nucleotide dissociation inhibitors (RhoGDIs) bind to Rho GTPases and control their spatiotemporal activity [8,9]. 

There are a large number of Rho GEFs and Rho GAPs, whereas the RhoGDI family only has three members in mammals: RhoGDI1 is ubiquitously expressed in various cells [10]; RhoGDI2 is preferentially expressed in hematopoietic cells [11,12]; and RhoGDI3 is expressed in the brain, testes, and pancreas [13,14]. RhoGDI1 and RhoGDI2 are present exclusively in the cytoplasm and form complexes with most Rho GTPases. In contrast, RhoGDI3 is associated with the Golgi complex and exhibits specificity for interactions with RhoB and RhoG [15]. In addition, little is known about the association between RhoGDI3 and human cancer. Therefore, we will focus on RhoGDI1 and RhoGDI2, but not RhoGDI3. RhoGDIs interact with most Rho GTPases in the cytoplasm and prevent Rho GTPases from binding to GEFs or their effector molecules. Thus, RhoGDIs have been considered to be negative regulators of Rho GTPases [16]. When Rho GTPases are dissociated from RhoGDIs, they can bind to the plasma membrane and be activated by GEFs [17]. The interaction between Rho GTPases and RhoGDIs is dynamically regulated by several mechanisms, including interactions with specific proteins or lipids, phosphorylation, ubiquitination, and sumoylation [18]. Accumulating evidence has shown that RhoGDIs are implicated in cancer cell migration, invasion, metastasis, and chemoresistance via the deregulation of the Rho GTPase signaling pathway [19,20], making them an attractive target for cancer treatment. Here, we review how RhoGDIs control the function of Rho GTPases by regulating their spatiotemporal activity and describe the regulatory mechanisms of the dissociation of Rho GTPases from RhoGDIs. We also discuss the role of RhoGDIs in cancer progression and their potential uses for therapeutic intervention.

## 2. Regulation of Rho GTPases by RhoGDIs

RhoGDIs contain a flexible *N*-terminal domain and a hydrophobic C-terminal domain. The N-terminal domain of RhoGDIs binds to switch I and switch II domains of Rho GTPases, which are the binding region for GEFs and GAPs. The interaction between these two domains inhibits the transition between the GTP- and GDP-bound forms [21,22,23]. The C-terminal domain of RhoGDIs forms a hydrophobic pocket and interacts with the membrane targeting prenyl moiety of Rho GTPases [24,25]. Most Rho GTPases bind to RhoGDIs in the cytoplasm and reside in their inactive form. When Rho GTPases are disengaged from RhoGDIs, they can integrate into the plasma membrane, where they are activated by RhoGEFs. Re-association of Rho GTPases with RhoGDIs mediates the extraction of Rho GTPases from the membrane and recycles them to the cytosol [26], as shown in Figure 1. Therefore, RhoGDIs were originally characterized as inhibitory regulators of Rho GTPases. Recent studies, however, have shown that they regulate Rho GTPases activity in more complex manners.

RhoGDIs can also act as chaperones for Rho GTPases. Active Rho GTPases are located on the plasma membrane, whereas most Rho GTPases are present in their inactive state in the cytosol [27,28]. The C-terminus of Rho GTPases is prenylated and thus highly hydrophobic. Although prenylation of Rho GTPases is required for their membrane targeting and signaling, free prenylated Rho GTPases in the cytosol are highly unstable. In fact, non-prenylated Rho GTPases fail to localize at the proper action site but they are more stable than prenylated Rho GTPases [29,30]. In addition, inactive Rho GTPases, which are not bound to RhoGDI1 in the cytosol, are degraded through proteasome-dependent mechanisms. RhoGDIs can bind to the prenyl moiety of Rho GTPases through a hydrophobic pocket at their C-terminus. This interaction stabilizes cytosolic Rho GTPases and protects them from proteasomal degradation [30,31]. Moreover, RhoGDIs not only extract Rho GTPases from membranes but also shuttle them to other proper membranes. Rho GTPases mutants, which cannot bind to RhoGDI1, fail to translocate to the plasma membrane. For example, Rac1 mutant (RacR66E), which cannot bind to RhoGDI1, failed to translocate to the membrane site in response to hepatocyte growth factor (HGF) stimulation [32]. Similarly, Cdc42 mutant (Cdc42R66A), which cannot interact with RhoGDI1, accumulated in the perinuclear membrane region [33]. RhoGDIs thus form soluble complexes with GDP-bound Rho GTPases in the cytosol and regulate the delivery and extraction of Rho GTPases to their action site. When RhoGTPases are released from RhoGDIs, they can bind into membranes and be activated by GEFs. Dissociation of Rho GTPases from RhoGDIs is necessary for membrane association and activation by GEFs. Re-association of RhoGDIs is also necessary for the recycling of Rho GTPases. These suggest that RhoGDIs are essential for correct targeting and regulation of Rho GTPase activity through regulation of spatial restriction as well as guidance and accessibility to regulators and effectors.

## 3. Regulation of RhoGDIs Activity

Association and disassociation of RhoGDIs with Rho GTPases are important for cytosol-membrane cycling of Rho GTPases, which serves a major role in regulating their activity and function. Interaction of RhoGDIs with selective Rho GTPases is regulated by a range of mechanisms: regulations by phospholipids; protein–protein interaction; and post-transcriptional modification (PTM), such as phosphorylation, sumoylation, acetylation, and ubiquitination. The detailed mechanisms of these regulations are discussed below.

### 3.1. Regulation of RhoGDIs by Phospholipids 

Several relevant lipids can regulate the binding of RhoGDIs to Rho GTPases. Phosphatidic acid and anionic phospholipids can promote partial release of Rho GTPases from RhoGDI1, which allows association of Rho GTPases with GEF proteins or other dissociation factors that completely release RhoGDI1 and activate Rho GTPases [34,35]. Furthermore, phosphoinositide promotes dissociation of RhoGDI1 from Rho GTPases indirectly through other mediators. For example, phosphatidylinositol 4,5-bisphosphate (PtdIns(4,5)P2) activates ezrin, radixin, and moesin (ERM) proteins, thereby promoting their interaction with RhoGDI1 and resulting in displacement of RhoGDI1 from RhoA [36,37]. Diacylglycerol kinase ζ (DGKζ) stimulates production of phosphatidic acid, thus activating PAK1 and resulting in the release of RhoGDI1 from Rac1 [38]. 

### 3.2. Regulation of RhoGDIs by Protein–Protein Interaction 

Displacement of RhoGDIs with Rho GTPases can be regulated through interaction with specific proteins, such as ERM proteins, FERM domain-containing protein 7 (FRMD7), the tyrosine kinase Etk, the p75 neurotrophin receptor, TROY (an orphan receptor of the tumor necrosis factor family), and EphrinB1. ERM proteins directly interact with RhoGDI1 and promote the release of various Rho GTPases family members, thereby activating them [39]. However, subsequent studies have shown that ERM proteins selectively control the release of RhoA from RhoGDI1 in vivo [36]. FRMD7 also binds to RhoGDI1 and specifically activates Rac1, thereby regulating reorganization of actin filaments and neuronal growth [40]. On the contrary, Etk interacts with RhoA but not Rac1 and Cdc42 and disrupts the interaction between RhoA and RhoGDI1 to promote membrane translocation and subsequent activation of RhoA in a pleckstrin homology domain-dependent manner [41]. The p75 neurotrophin receptor and TROY can interact with RhoGDI1. Furthermore, the interaction of RhoGDI1 and these proteins is enhanced by their ligands, myelin-associated glycoprotein (MAG), and neurite outgrowth inhibitor (NOGO), thereby resulting in activation of RhoA by promoting its release from RhoGDI1 [42,43]. Similarly, a recent study found that RhoGDI1 can directly interact with ephrinB1, which is known to regulate cell-to-cell adhesion and cell migration in various biological processes [44]. Engagement of EphB2 receptor with ephrinB1 enhances its interaction with RhoGDI1 to promote the release of RhoA. Interestingly, ephrinB1 promotes the release of RhoA and Rac1 but activates only RhoA, suggesting that the release of Rho GTPases from RhoGDI1 by ephrinB1 signaling is not sufficient for Rho GTPase activation but contributes to activating specific Rho GTPases. Therefore, additional factors, such as specific RhoGEFs and displacement of RhoGDIs, may be required for ephrinB1-mediated RhoA activation. The interaction between RhoGDI and protein does not always promote the dissociation of RhoGDI and Rho GTPases. In fact, a recent report has shown that Wilms tumor gene on the X chromosome (WTX)—a putative tumor suppressor gene—inhibits CDC42 activation through binding to RhoGDI2 and then stabilizing the complex formation of RhoGDI1 and CDC42 [45].

### 3.3. Regulation of RhoGDIs by Post-Translational Modifications 

Dissociation of Rho GTPases from RhoGDIs is a vital step in their activation. The interaction between Rho GTPases and RhoGDIs can be regulated by post-translational modifications, such as phosphorylation, sumoylation, acetylation, and oxidation, as shown in Table 1. Generally, phosphorylation of RhoGDIs decreases their affinity for Rho GTPases, thereby promoting the release of Rho GTPases and their subsequent activation by RhoGEFs [5,17]. For instance, phosphorylation of RhoGDI1 by protein kinases, such as Src, PKCα, P21 activated kinase 1 (PAK1), and FER, promotes the release of specific Rho GTPases and their subsequent spatiotemporal activation [46,47,48,49,50]. Src kinase phosphorylates RhoGDI1 at Tyr156 to promote translocation of RhoGDI1 to the plasma membrane and local activation of RhoA, Rac1, and Cdc42 [46]. β8 integrin-bound protein tyrosine phosphatase-PEST dephosphorylates RhoGDI1 and releases it from the membrane, which can terminate Rac1 and Cdc42 signaling at the leading edge of migrating cells [51]. RhoGDI1 phosphorylation at Ser101 and Ser174 by PAK1 promotes dissociation and activation of Rac1 but not RhoA [49]. Interestingly, because PAK1 is activated by Rac1 and Cdc42 to regulate cell migration [52], there may be positive feedback between PAK1 activation and RhoGDI1 phosphorylation for Rac1 signaling. However, phosphorylation of RhoGDIs does not always induce the release of Rho GTPases. Cyclic AMP-dependent protein kinase A (PKA) can phosphorylate RhoGDI1 at Ser174 and RhoA at Ser188, which increases the affinity of RhoGDI1 to RhoA, leading to inhibition of RhoA signaling [53,54]. In contrast, a recent study reported that 14-3-3τ interacts with phosphorylated RhoGDI1 at Ser174 by epidermal growth factor (EGF) stimulation, resulting in the release and subsequent activation of RhoA, Rac1, and Cdc42 [55]. In addition, protein phosphatase 1B dephosphorylates RhoGDI1 to decrease RhoGDI1 interaction with 14-3-3τ and Rho GTPase activation, thus suppressing EGF-induced breast cancer cell migration [56]. Therefore, reversible phosphorylation of RhoGDIs by a kinase/phosphatase could constitute a critical mechanism to precisely regulate the spatiotemporal activation of Rho GTPases.

In addition to phosphorylation, sumoylation is reversible PTM in which a small ubiquitin-like modifier (SUMO) moiety is covalently linked to lysine residues in the target proteins. This PTM regulates various biological functions, including transcription, DNA repair, protein interaction, stability, and localization [57,58]. RhoGDI1 was reported to be SUMOylated at Lys138, which increases its binding affinity for Rho GTPases. Thus, sumoylation of RhoGDI1 negatively regulates the activity of RhoA and Rac1, leading to a decrease in cancer cell migration. X-linked inhibitor of apoptosis (XIAP) binds to RhoGDI1 via its RING domain and inhibits RhoGDI1 sumoylation [59,60]. Gene related to anergy in lymphocytes (GRAIL), a ring finger ubiquitin E3 ligase, also binds to and ubiquitinates RhoGDI1. Interestingly, ubiquitination of RhoGDI1 by GRAIL does not promote proteasomal degradation but results in specific inhibition of RhoA through undefined mechanisms [61]. Recent studies have reported that RhoGDI1 can be acetylated at Lys127 and Lys141 by CBP, p300, and pCAF, and can be de-acetylated by Sirt2 and HDAC6. Acetylation of RhoGDI1 decreases its binding affinity for RhoA, thereby promoting the activation of RhoA [62,63]. More recently, it has been reported that RhoGDI1 oxidation by hydrogen peroxide promotes the release of RhoA and thus increases its activity [64]. Many researchers have focused on RhoGDI1 phosphorylation in the regulation of Rho GTPases activity, whereas the mechanism of other PTMs has not been studied much. The molecular mechanism of the sumoylation, acetylation, oxidation, and phosphorylation of RhoGDI1 is an unresolved but interesting area of research.

## 4. Functions of RhoGDIs in Human Cancers

It was originally reported that RhoGDI1 is ubiquitously expressed in all mammalian organs, whereas RhoGDI2 is expressed in hematopoietic cells. However, accumulating evidence has shown that the expression of RhoGDI1 and RhoGDI2 is upregulated or downregulated in different human cancers and closely associated with cancer malignancy, as shown in Table 2, indicating that the functions of RhoGDIs in cancer progression are complex and controversial. For instance, upregulated expression level of RhoGDI1 is observed in colorectal cancers and involved in enhanced invasion and chemoresistance [65,66]. RhoGDI1 expression is also upregulated in hepatocellular carcinoma (HCC) cell lines and tissues with highly metastatic potential. In this case, RhoGDI1 activates the PI3K and MAPK pathway by increasing RhoA expression, thereby promoting cancer cell proliferation and migration [67]. In contrast, RhoGDI1 expression is downregulated in lung and brain cancers, and negatively correlated with the degree of malignancy [68,69], indicating that RhoGDI1 functions as a metastasis suppressor. Conflicting results of RhoGDI1 on tumor progression have also been observed in breast cancers. The expression levels of RhoGDI1 are upregulated or downregulated in breast cancer tissues in different studies [70,71]. RhoGDI1 protects cancer cells from drug-induced apoptosis. Overexpression of cellular RhoGDI1 increased resistance to etoposide- and doxorubicin-induced apoptosis in breast and lymphoma cells, and depletion of RhoGDI1 had the opposite effect. The anti-apoptotic activity of RhoGDI1 may be derived from its ability to inhibit caspase-3-mediated cleavage of Rac1 [72]. These dual functions of RhoGDI1, i.e., the tumor-promoting and -suppressing functions, may be due to differences in cellular components that can interact with RhoGDI1 between different cancer cell types. For instance, RhoGDI1 suppresses the migration of estrogen receptor (ER)-positive MCF-7 breast cancer cell line but does not regulate the migration of ER-negative MDA-MB-231 cell line [73,74]. RhoGDI2 is also differentially expressed in human cancers. The expression level of RhoGDI2 is downregulated in Hodgkin’s lymphoma and bladder cancer, and reduced RhoGDI2 expression is correlated with the invasive and metastatic phenotypes [75,76]. RhoGDI2 inhibits Rac1 activation by osteopontin secreted by macrophage in the tumor microenvironment, and thus suppresses metastasis of bladder cancer cells [77]. However, many studies have reported that the expression levels of RhoGDI2 are upregulated in various human cancers, such as ovarian and gastric cancers, and correlated with advanced tumor stage and increased metastatic potential of tumor cells [78,79,80]. Moreover, lymph node metastasis-positive gastric tumors have higher levels of RhoGDI2 expression than lymph node metastasis-negative tumors [80]. RhoGDI2 also protects gastric cancer cells from various chemotherapeutic agents, such as etoposide, staurosporine, and cisplatin, through upregulation of Bcl-2 expression [81,82]. Although RhoGDI1 is completely resistant to degradation during apoptosis, RhoGDI2 is well-characterized as a substrate for caspases and cleaved during apoptosis in various cells [72]. Therefore, RhoGDI2 may act as an antiapoptotic molecule via a distinct mechanism from RhoGDI1, and may do so prior to caspase activation during drug-induced apoptosis. All these studies indicate that RhoGDIs can function as both a tumor suppressor and tumor promoter. RhoGDIs play a role in cancer progression by regulating activity of Rho GTPase family members, and their regulators are very diverse in different cell types. Thus, the opposite effects of RhoGDIs may be due to diversities in the genetic backgrounds of different cancer cell types or tumor stages.

RhoGDIs are differentially expressed in many human cancers, and their expression is closely involved in the malignant tumor phenotypes. They play functionally significant roles in cancer malignancy by regulating the activity of Rho GTPases at the appropriate time and place. RhoGDI1 expression can be regulated both transcriptionally and post-transcriptionally. A recent study has described that RhoGDI1 is a direct transcriptional target of activation transcription factor 4 (ATF4), which is implicated in cell migration and tumor metastasis [83,84]. RhoGDI1 expression is also post-transcriptionally regulated by microRNA (miR)-151, which is expressed together with its host gene focal adhesion kinase (FAK). During hepatocellular tumor progression, miR-151 functions synergistically with FAK to enhance the invasion and metastasis of hepatocellular carcinoma cells by promoting the downregulation of RhoGDI1 [85]. miR-25 also exerts its pro-metastatic function by directly targeting RhoGDI1. Downregulation of RhoGDI1 by miR-151-5p/miR-25 enhances the expression of Snail, thereby promoting the migration and invasion of hepatocellular carcinoma cells [86]. RhoGDI1 mRNA binds to the RNA-binding protein PCBP2, which is an essential regulator of miRNA biogenesis, stability, and activity. The interaction between PCBP2 and the 3′UTR of RhoGDI1 mRNA induces a local change in RNA structure that favors interaction with miR-151-5p/miR-16, thus leading to the suppression of RhoGDI1 expression, which promotes glioma cell migration and invasion [87]. RhoGDI1 expression can be also regulated by the ubiquitin–proteasome system. A recent study reported that FBW E3 ubiquitin ligase was able to bind to and degrade RhoGDI1 in a proteasome-dependent manner, promoting bladder cancer cell migration [88]. Moreover, angiotensin II regulates stability of RhoGDI1 and RhoGDI2 by SUMOylation and ubiquitination via AT1 receptor activation and thus affects vascular smooth muscle proliferation and vascular remodeling [89]. We anticipate that other factors controlling the expression of RhoGDIs at the transcriptional and post-transcriptional levels will be elucidated. Some of these may be related to cancer malignancy by modulating the expression level of RhoGDIs.

## 5. Conclusions and Future Directions

Although the roles of RhoGDIs in regulating the activity of Rho GTPases have been established over the past two decades, recent findings have described the rather complicated mechanisms of this regulation. In this review, we have discussed how RhoGDIs regulate Rho GTPase stability and correct Rho GTPase targeting and activation, as well as how several factors, such as phospholipids, interacting proteins, and PTMs, regulate the release of Rho GTPases from RhoGDIs. Although many studies have focused on identifying the factors that regulate the interaction between these two proteins, the details of this mechanism is still unclear. The GDP-GTP cycling of each Rho GTPase is highly and dynamically regulated during cancer cell migration. As discussed in this review, PTMs, including phosphorylation, sumoylation, ubiquitination, acetylation, and oxidation, are highly dynamic cellular processes. Therefore, co-ordination of PTMs is necessary for the spatiotemporal regulation of Rho GTPases in individual cells in response to different stimuli. Thus, exploring additional factors regulating the affinity of RhoGDIs and specific Rho GTPases will be useful for understanding the precise mechanisms controlling the dynamic activity of Rho GTPases.

The expression of RhoGDIs is deregulated in many different tumors and associated with tumor malignancy, including migration, invasion, metastasis, and chemoresistance [19,20]. RhoGDIs appear to play significant roles through correct membrane-targeting of Rho GTPases and regulation of their activity [27,28]. Further studies on the detailed molecular mechanisms regarding how RhoGDIs control the activity of Rho GTPases in various human cancers are needed to define the significance of RhoGDIs in cancer progression. In addition, these studies should consider the interplay between RhoGDIs and other factors, such as phospholipids, interacting proteins, and PTMs, in a combined manner. Given the significant contribution of RhoGDIs in tumor malignancy, they may be a strong candidate therapeutic targets for inhibiting Rho GTPases activity, which is exhibited in many human cancers [17,18]. Both competitive inhibitors that can inhibit the initial targeting of Rho GTPases and interfacial inhibitors that can suppress the release of RhoGDIs from Rho GTPases can be potential strategies for cancer treatment. Gene silencing of RhoGDIs using RNA interference may also be a useful strategy for treating cancers, in which they function as a tumor promoter. However, attention should be paid to targeting these pathways. The expression pattern of RhoGDIs in human cancers clearly shows that a universal and simple strategy for targeting RhoGDIs is not effective against all types of cancers. Successful treatment in one type of cancer can be very harmful in other types of cancers.

RhoGDIs can function as both a tumor suppressor and tumor promoter, depending on the tumor type and microenvironment. Extensive expression profiles of RhoGDIs in various human cancers can help establish whether they function as tumor suppressors or tumor promoters in different tumor types and stages. Additional studies are needed to define the tumor-specific functions of RhoGDIs in various human cancers. In addition, future studies should focus on resolving the interconnected role of Rho GTPases and RhoGDIs in cancer progression to allow the development of effective therapeutic strategies against these targets. These efforts will provide improved cancer therapies and an in-depth understanding of tumor biology.

## Figures and Tables

**Figure 1 cells-08-01037-f001:**
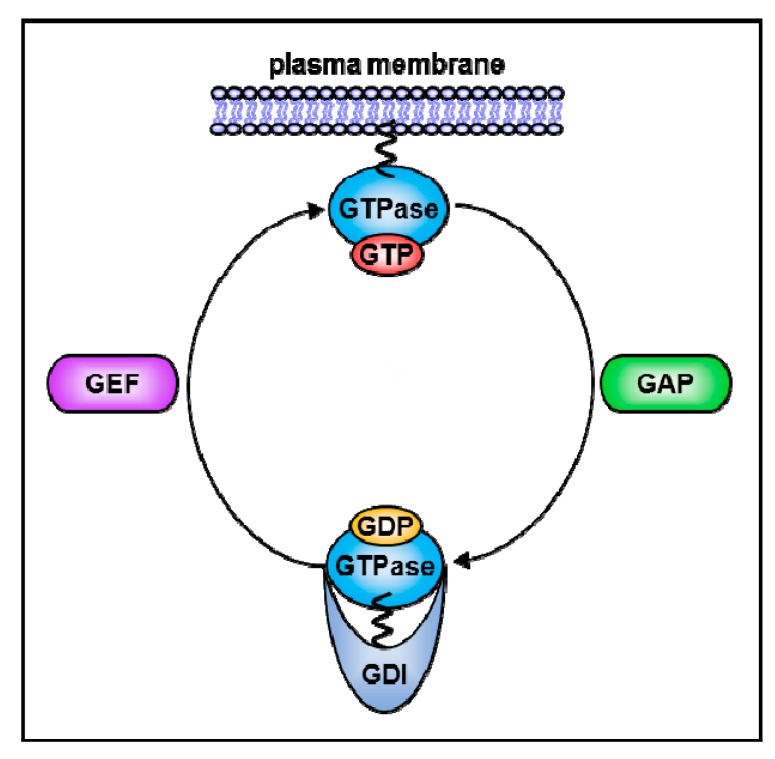
The regulation of Rho GTPases by GEPs, GAPs, and GDIs. GEFs bind to GDP-bound RhoGTPases and promotes the exchange of GDP for GTP, thereby activating RhoGTPases. GAPs bind to GTP-bound RhoGTPases and catalyze the exchange of GDP for GTP, thereby inactivating RhoGTPases. The N-terminal domain of RhoGDIs binds to switch I and II domains of RhoGTPases. The C-terminal region of RhoGDIs forms a hydrophobic pocket and binds to prenylated RhoGTPases. Therefore, RhoGDIs can extract RhoGTPases from plasm membrane by binding the isoprenoid moiety and sequester them away in the cytoplasmic compartment.

**Table 1 cells-08-01037-t001:** Regulation of Rho GTPases activity by post-translational modifications of Rho GDP dissociation inhibitor (RhoGDI).

PTMs	Regulator	Target Site	Effects	Refs.
Phosphorylation	Src PAK1 PKCα PKA FER	Tyr27, Tyr156 Ser101, Ser174 Ser34, Ser96 Ser174 ND	Promotes release of RhoA, Rac1, and Cdc42 Promotes release of Rac1 Promotes release of RhoA and RhoG Induces the complex formation with RhoA Prevents the interaction with Rac1	[45,47,48,49,50,53]
Dephosphorylation	PTP-PEST PPM1B	Tyr156 Ser174	Inactivation of Rac1 and Cdc42 Inactivation of RhoA, Rac1, and Cdc42	[51,56]
Sumoylation	ND	Lys138	Increases the binding to RhoA, Rac1 and Cdc42	[59]
Ubiquitination	GRAIL	ND	Inhibits RhoA activation	[61]
Acetylation	p300, pCAF	Lys127, Lys141	Promotes release of RhoA	[62,63]
Oxidation	ND	ND	Promotes RhoA activation	[64]

ND, not determined.

**Table 2 cells-08-01037-t002:** Expression of RhoGDI1 and RhoGDI2 in human tumor tissues.

RhoGDIs	Tumor	RNA/Protein	Expression	Correlation	Refs.
RhoGDI1	Colorectal cancer Hepatocellular carcinoma Lung adenocarcinoma Brain cancer Breast cancer	Protein Protein RNA RNA Protein RNA, Protein	Up Up Down Down Down Up	St, M M M ND M ND	[65,66,67,68,69,70,71]
RhoGDI2	Hodgkin’s lymphoma Bladder cancer Ovarian cancer Gastric cancer Breast cancer	Protein RNA, Protein RNA Protein Protein	Down Down Up Up Up	A M St St, M M	[75,76,77,78,79,80,82]

St, stage; M, metastasis; A, apoptosis; ND, not determined.

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
