# Peer review of "Regulation of Rho GTPases by RhoGDIs in Human Cancers"

_cells, 2019, doi:10.3390/cells8091037_

Round 1

Reviewer 1 Report

The review addresses a gap in the literature as relatively few reviews on RhoGDIs have been published in the past 5 years.

The review provides good definition of the functions of RhoGDIs and levels of RhoGDI regulation through post-translational modifications.

There are several issues that need to be addressed.

The authors state in the abstract “In this review, we summarize recent studies that described the cellular functions of RhoGDIs and their mechanisms in regulating cancer progression. We also discuss the mechanistic implications in the design and application of RhoGDI-targeting strategies for future cancer therapies.” However, the actual review falls short of these promises.

Section 4 on functions of RhoGDIs in cancers is disorganized and does not provide a clear picture of the role of aberrant regulation of these proteins in cancer. Perhaps a table of cancers with changes in expression (indicating protein or RNA) and whether tumor tissues or cell lines were tested would provide some insights and provide a reader with the opportunity to critically evaluate the state of the literature.

There is only superficial discussion on how or why changes in expression or post-translational modification of RhoGDIs would lead to alteration of cancer relevant functions. For example, no discussion on mechanisms of RhoGDI impact on chemoresistance is provided, although at several places in the text RhoGDIs are stated as important in chemoresistance.

Discussion of miRs in regulating RhoGDIs should be consolidated and possibly placed in Section 3 as another mechanism of regulating activity/expression. This is an important and emerging area of RhoGDI regulation that deserves a more considered discussion.

There is no meaningful discussion on RhoGDI targeting strategies for future therapeutics although this is listed as a review topic in the abstract and introduction. What would be possible targeting strategies? What are the outcomes of existing knock down studies in tumor cells that might provide insights into the outcomes of targeting RhoGDIs? What are the implications of the conflicting evidence for the impact of RhoGDI expression in certain tumors on design and implementation of therapeutic targeting?  

Authors should carefully review English language usage.

Authors should carefully review the accuracy of citations. For example, reference 75 does not relate to colon, pancreatic or ovarian cancer.

Author Response

The review addresses a gap in the literature as relatively few reviews on RhoGDIs have been published in the past 5 years.

The review provides good definition of the functions of RhoGDIs and levels of RhoGDI regulation through post-translational modifications.

There are several issues that need to be addressed.

1. The authors state in the abstract “In this review, we summarize recent studies that described the cellular functions of RhoGDIs and their mechanisms in regulating cancer progression. We also discuss the mechanistic implications in the design and application of RhoGDI-targeting strategies for future cancer therapies.” However, the actual review falls short of these promises.

Response: Thank you for valuable suggestion. We have now modified the abstract (page 2. line 36 ~ 39) and more discussed potential uses for therapeutic intervention (page 11. line 270 ~ 284).

2. Section 4 on functions of RhoGDIs in cancers is disorganized and does not provide a clear picture of the role of aberrant regulation of these proteins in cancer. Perhaps a table of cancers with changes in expression (indicating protein or RNA) and whether tumor tissues or cell lines were tested would provide some insights and provide a reader with the opportunity to critically evaluate the state of the literature.

Response: The reviewer makes an excellent point. We have now added a table describing the expression of RhoGDIs and their role in human cancers. Please see table 2 on page 9.

3. There is only superficial discussion on how or why changes in expression or post-translational modification of RhoGDIs would lead to alteration of cancer relevant functions. For example, no discussion on mechanisms of RhoGDI impact on chemoresistance is provided, although at several places in the text RhoGDIs are stated as important in chemoresistance.

Response: According to reviewer’s suggestion, we have now discussed mechanisms how RhoGDIs regulate chemoresistance of cancer cells on page 8, line 212 ~ 215 and page 9, line 224 ~ 230.

4. Discussion of miRs in regulating RhoGDIs should be consolidated and possibly placed in Section 3 as another mechanism of regulating activity/expression. This is an important and emerging area of RhoGDI regulation that deserves a more considered discussion.

Response: As reviewer suggested, we have more discussed the regulation of RhoGDI1 expression by miRNA on new paragraph in Section 3 (page 10, line 241 ~ 253).

5. There is no meaningful discussion on RhoGDI targeting strategies for future therapeutics although this is listed as a review topic in the abstract and introduction. What would be possible targeting strategies? What are the outcomes of existing knock down studies in tumor cells that might provide insights into the outcomes of targeting RhoGDIs? What are the implications of the conflicting evidence for the impact of RhoGDI expression in certain tumors on design and implementation of therapeutic targeting?  

Response: We are grateful to the reviewer for these comments, and we have now more discussed strategies targeting RhoGDI1 for therapeutic intervention on page 11, line 271 ~ 292.

6. Authors should carefully review English language usage.

Response: The manuscript has been corrected by a native speaker of Editage, a professional English proofing company. We have now attached the certificate of editing.

7. Authors should carefully review the accuracy of citations. For example, reference 75 does not relate to colon, pancreatic or ovarian cancer.

Response: We thank the reviewer for making us aware our error. We have now corrected the mistake and carefully checked if there was another mistake.

Reviewer 2 Report

The manuscript can be greatly strengthened by critically evaluating the expression and alternations of GDIs in human cancers using published and public databases. The evaluations can help to validate some observations regarding altered GDIs expression in cancer cited in the manuscript. 

The manuscript needs to professionally edited. 

Author Response

The manuscript can be greatly strengthened by critically evaluating the expression and alternations of GDIs in human cancers using published and public databases. The evaluations can help to validate some observations regarding altered GDIs expression in cancer cited in the manuscript. 

The manuscript needs to professionally edited. 

Response: We appreciate the reviewer’s remarks. The manuscript has been edited twice by English language editing service Editage Inc (https://www.editage.co.kr). The certificate of editing is attached.

We thank all the reviewers for their insightful and thorough comments and suggestions. We believe this reorganization would clarify the subject of this manuscript and improve readability. We look forward to a positive assessment of this revised manuscript.

Round 2

Reviewer 1 Report

The authors were generally responsive to the previous critique, but some substantial concerns remain.

Major concerns:

Because there are few recent reviews on RhoGDIs in the literature, this manuscript has the potential to make a significant contribution. However, the numerous omissions of more recent and pertinent published findings diminishes the impact of the current review. There are important publications within the past 3 years on RhoGDIs in tumors, PTMs, lipid regulation and other topics discussed by the authors that are not included in this review. Citations that are up to date are an expectation for a high quality review article. The inclusion of data on tumor expression and alterations from public databases as suggested in the prior review is missing and weakens the document. Therapeutic potential, although highighted in the abstract and introduction, is only superficially addressed. This section is expanded compared to the first version, but the inserted text (lines 256-271) does not include any direct evidence or a single citation to support the statements. Similarly, the paragraph starting on line 272 states that RhoGDIs have opposite effects on cancer progression depending on the tumor type, but this claim is poorly supported.

Minor comments:

Line 46--stated that RhoGDI3 acts in a different way--need to define in what way since this is the basis of excluding further disucssion of RhoGDI3. Line 86-what mutations disrupt interaction between RhoGDIs and Rac/Cdc42? Line 119--The cited paper exclusively involves studies in cells which the authors define as "in vivo", but in the context of the review this statement is misleading.

Author Response

The authors were generally responsive to the previous critique, but some substantial concerns remain.

Major concerns:

Because there are few recent reviews on RhoGDIs in the literature, this manuscript has the potential to make a significant contribution.

However, the numerous omissions of more recent and pertinent published findings diminishes the impact of the current review. There are important publications within the past 3 years on RhoGDIs in tumors, PTMs, lipid regulation and other topics discussed by the authors that are not included in this review. Citations that are up to date are an expectation for a high quality review article.

Response: We thank the reviewer for construct comments. According to the reviewer suggestion, we have added recent publications over the past three years related to the expression and function of RhoGDIs in cancer and PTMs.

Zhu, G.F.; Xu, Y.W.; Li, J.; Niu, H.L.; Ma, W.X.; Xu, J.; Zhou, P.R.; Liu, X.; Ye, D.L.; Liu, X.R.; Yan, T.; Zhai, W.K.; Xu, Z.J.; Liu, C.; Wang, L.; Wang, H.; Luo, J.M.; Liu, L.; Li, X.Q.; Guo, S.; Jiang, H.P.; Shen, P.; Lin, H.K.; Yu, D.H.; Ding, Y.Q.; Zhang, Q.L. Mir20a/106a-WTX axis regulates RhoGDIa/CDC42 signaling and colon cancer progression. Nat Commun. 2019, 10:112.

(Please see line 133~137)

Huang, D.; Lu, W.; Zou, S.; Wang, H.; Jiang, Y.; Zhang, X.; Li, P.; Songyang, Z.; Wang, L.; Wang, J.; Huang, J.; Fang, L. Rho GDP-dissociation inhibitor α is a potential prognostic biomarker and controls telomere regulation in colorectal cancer. Cancer Sci. 2017, 108:1293-1302

(line 186~189)

Ahmed, M.; Sottnik, J.L.; Dancik, G.M.; Sahu, D.; Hansel, D.E.; Theodorescu, D.; Schwartz, M.A. An Osteopontin/CD44 Axis in RhoGDI2-Mediated Metastasis Suppression. Cancer Cell. 2016, 30:432-443.

(line 207~209)

Pasini, S.; Liu, J.; Corona, C.; Peze-Heidsieck, E.; Shelanski, M.; Greene, L.A. Activating Transcription Factor 4 (ATF4) modulates Rho GTPase levels and function via regulation of RhoGDIα. Sci Rep. 2016, 6:36952

(line 227~230)

Zeng, P.; Sun, S.; Li, R.; Xiao, Z.X.; Chen, H. HER2 Upregulates ATF4 to Promote Cell Migration via Activation of ZEB1 and Downregulation of E-Cadherin. Int J Mol Sci. 2019 6;20(9).

(line 227~230)

Zhu, J.; Li, Y.; Chen, C.; Ma, J.; Sun, W.; Tian, Z.; Li, J.; Xu, J.; Liu, C.S.; Zhang, D.; Huang, C.; Huang, H. NF-κB p65 Overexpression Promotes Bladder Cancer Cell Migration via FBW7-Mediated Degradation of RhoGDIα Protein. Neoplasia. 2017, 19:672-683

(line 241~244)

Dai, F.; Qi, Y.; Guan, W.; Meng, G.; Liu, Z.; Zhang, T.; Yao, W. RhoGDI stability is regulated by SUMOylation and ubiquitination via the AT1 receptor and participates in Ang II-induced smooth muscle proliferation and vascular remodeling. Atherosclerosis. 2019, 288:124-136

(line 244~246)

The inclusion of data on tumor expression and alterations from public databases as suggested in the prior review is missing and weakens the document.

Response: According to reviewer’s suggestion, we have now added a modified table describing the expression of RhoGDI (indicating protein or RNA) in human cancer tissues in more detail. (Page 10, table 2)

Therapeutic potential, although higlighted in the abstract and introduction, is only superficially addressed. This section is expanded compared to the first version, but the inserted text (lines 256-271) does not include any direct evidence or a single citation to support the statements.

Similarly, the paragraph starting on line 272 states that RhoGDIs have opposite effects on cancer progression depending on the tumor type, but this claim is poorly supported.

Response: As the reviewer suggested, we have now cited references to support our statements (line279~287). References to dual function of RhoGDIs on cancer malignant phenotypes have already been addressed in section “function of RhoGDIs on human cancer”. To clarify this, we have modified the sentence (line 294~295)

Minor comments:

Line 46--stated that RhoGDI3 acts in a different way--need to define in what way since this is the basis of excluding further disucssion of RhoGDI3.

Response: We have more discussed some of the different ways in which RhoGDI1/2 and RhoGDI3 function. (Line 43~47)

Line 86-what mutations disrupt interaction between RhoGDIs and Rac/Cdc42?

Response: We have now added Rac1 substitution mutant (RacR66E) and Cdc42 substitution mutant (Cdc42R66A). Please see line 85 and 86

Line 119--The cited paper exclusively involves studies in cells which the authors define as "in vivo", but in the context of the review this statement is misleading.

Response: As the reviewer suggested, we have removed “in vivo”.

Reviewer 2 Report

Although the revisions did not address the suggestions to strengthen the manuscript through bio-informatic data, the review manuscript by itself is adequate.

Author Response

Although the revisions did not address the suggestions to strengthen the manuscript through bio-informatic data, the review manuscript by itself is adequate.

Response: We thank the reviewer for the positive and constructive comments. We did not use bio-informatic data because informatic data like TCGA is not commonly used in review papers. Instead, we have now added a modified table describing the expression of RhoGDI (indicating protein or RNA) in human cancer tissues in more detail. (Page 6, table 2).